# Comparison of sequential data analysis and functional data analysis for locomotor adaptation

**Torin Quinlivan**[1]*, **Kacy Kane**[2], **Christopher M. Hill**[3], **Duchwan Ryu**[2]

**1** Department of Mathematics, Knox College, Galesburg, Illinois, United States of America, **2** Department of Statistics and Actuarial Science, Northern Illinois University, DeKalb, Illinois, United States of America, **3** School of Kinesiology, Louisiana State University, Baton Rouge, Louisiana, United States of America

\* torin.quinlivan@gmail.com

**Data availability statement:** All relevant data are within the paper and its Supporting information files.

## Abstract

Learning rates for skills such as walking may depend on circumstances or time, while incentivization with punishments or rewards may affect human skill learning. We consider a state space model for dynamically changed learning rates and figure out the effect of incentivization on the learning rates by utilizing a dynamically weighted particle filter. However, estimations of model parameters, including the learning rate, require a demanding computational burden, especially when the data are collected over a long period. To overcome computational difficulty, we utilize an efficient sequential Monte Carlo method, dynamically weighted particle filter, in the estimations of model parameters. Alternatively, we consider a functional data analysis for the learning rates and the effect of the incentivization. Two approaches have led to reasonable estimations of learning rates. We present the estimated learning rates and the effect of incentivization on the learning rates from two approaches, as well as the comparisons of their results.

## 1 Introduction

Walking is one of the many skills that we learn through a trial-and-error practice. To generate a new walking pattern, the brain must learn to correct movement errors based on feedback received from the body and environment. This adaptive process results in the learning and retention of a diverse repertoire of locomotor patterns that are maintained throughout our lifespan [1]. The plastic nature of walking adaptation serves as the basis for walking rehabilitation for neurological disease and injury, for example stroke [2,3]. Walking adaptation stems from the improved correction of errors by the central nervous system. There are two types of errors. The sensory prediction errors (SPE) result from discrepancies between expected sensory feedback such as visual, vestibular and proprioception and actual sensory feedback derived from our movement [4]. The reward prediction errors (RPE) is similar, but instead of sensory feedback, it results from differences in the reinforcement expected and the reinforcement received after a movement. Minimizing the discrepancies between one or both of these error types develops an efficient motor plan to perform the new walking pattern [2,3]. Interestingly, the type of feedback received during learning can differently influence learning

**Funding:** The author(s) received no specific funding for this work.

**Competing interests:** There are no competing interests.

rates. For instance, provided visual feedback of performance has been widely shown to enhance the rate of visuomotor learning. Others have utilized reinforcement feedback such as reward and punishment [5]. These types of feedback elicit different influences on skill learning. Most researchers have found that punishment feedback increases the rate of visuomotor adaptation and reward enhances motor memory of task [5–10]. Conversely, other tasks have noted increased learning with reward and visual feedback compared to punishment feedback. Interestingly, in the area of locomotor adaptation, studies have found mixed effects of reinforcement. For instance, one study found that both reward and punishment impaired locomotor adaptation [11], while another found that punishment increased memory and reward impaired adaptation [12]. Thus, the effect of reinforcement on locomotor learning are complex, multifaceted, and in need of a consensus and more robust analysis to understand the optimal conditions for reinforcement feedback usage.

Previous investigations have leveraged standard state space models to investigate retention factors and the learning rates in motor adaptation [13–15]. Using the movement kinematics collected from individual participants over time, a state space model can be used to estimate the retention and learning factors over the course of skill learning. Such analysis has provided interesting insights into how reward and punishment alter skill learning and change motor memory[9,13–15]. In this model, learning rate (i.e., parameter $\mathbb{B}$) refers to how fast the person adapts their movement to meet the environmental demand. Retention (parameter $\mathbb{A}$) refers to the ability to carry over or remember previous performed movement. Each of these constructs tell us something different about how the brain adapts and recalls our movement. More specifically, parameter $\mathbb{B}$ reflects the rapid adjustment of cerebellar circuits based on the large amount of sensory prediction errors present early in adaptation. Parameter $\mathbb{A}$ reflects how the movement pattern is stored in the neural connections between the motor cortex and the cerebellum. In summation, the parameters provide us with a picture of the underlying mechanisms used by the nervous system to facilitate adaptation across a short timescale [9,13–15]. Recent work has outlined the challenges associated with state space modeling in the context of skill learning. For instance, estimating retention factors is increasingly difficult with state space models, which may be subject to hidden factors that underlie motor memory [16]. Moreover, processes of learning are sources of stochasticity that are typically not taken into consideration by these models. Other practical factors also play a role. Missing values or unexpected and dynamic changes of movement patterns over time may limit the robust nature of state space modeling. Taken together, there is great need to develop new approaches to modeling robust motor learning behavior. Such models could pave the way for more accurate prediction of rehabilitation outcomes.

In this paper, we consider two approaches to overcome these challenges and to more accurately estimate the retention factors and learning rate of participants during a locomotor adaptation paradigm, where participants learn a new knee flexion pattern with visual feedback of their angular kinematics, and, moreover, examine the effect of reinforcement (i.e. reward and punishment) on these rates. The first approach constructs a state space model for the knee angles with group means and variances. Using a dynamically weighted particle filter [17,18], the parameters involved in the model are efficiently estimated. Based on the sequences of the estimated means and variances and the observed means and variances for each group of participants, we solve the structural equation model to estimate the retention factors and the learning rate. The second approach utilizes a Bayesian functional data analysis. The functional data analysis [19,20] handles a sequence of time-varying measurements as the realizations of a smooth function. We consider a more flexible nonlinear state space model and fit Bayesian cubic regression splines [21,22] to each of the group means of knee angles over time. The derivative of fitted curves can be used for the estimation of the retention rates,

learning rates, and their confidence intervals for inference. However, it should be noted that the confidence intervals of the derivative of estimated curves are not readily achievable. By the virtue of the Markov Chain Monte Carlo samples of the fitted curves, we derive the non-constant retention factors and learning rates and their confidence intervals. Furthermore, we investigate the effect of reward-punishment condition on the retention factors and learning rates and compare two approaches.

The rest of the paper is structured as follows. Sect 2 describes how the data was collected. Sect 3 describes the two methods in detail, with explanations of the parameters and full prior and posterior distributions, with Sect 3.1 detailing the state space model and Sect 3.2 detailing the functional data analysis. Sect 4 outlines the results of fitting the data using these methods and the parameter estimates. Finally, Sect 5 gives our conclusions.

## 2 Data collection methods

All procedures were approved by the Northern Illinois University Institutional Review Board [protocol number HS21-0399]. Thirty-three young healthy adults participated in this study [age range: 19–34 years, mean age $\pm$ standard deviation (SD): 24.61 $\pm$ 3.97 years, body height: 170.66 $\pm$ 34.32cm, body weight: 77.28 $\pm$ 15.57kg, males: 16, females: 17)]. Participants were classified as right-handed using the Edinburgh Handedness Inventory (EHI) (>50 implies right-handedness, mean handedness score $\pm$ SD: 94.74 $\pm$ 10.40) and were free of major physiological (musculoskeletal, neurological, cardiovascular) and psychological (drug abuse, depression, generalized anxiety) disorders. Participants were recruited from the local population of the University and the surrounding communities using word of mouth, electronic announcements, and posted flyers. Each participant was randomly allocated to one of three feedback groups: Reward ($n$ = 11), Punishment ($n$ = 11), Control ($n$ = 11). All participants were informed of the task goals by being read aloud a script before the start of the experiment. Each participant adapted to a new walking pattern on a treadmill (Woodway, Waukesha, WI) with a different type of visual feedback in two distinct task conditions. Particpants were read aloud task a script describing different conditions, the task goal of changing their knee flexion angle, and the meaning of the visual feedback. Afterwards, participants were asked to recall the task goals to ensure understanding of the procedures. Questions from the participants concerning procedures and goals were addressed by the experimenters. To begin the protocol, participants began walking in a dimly lit room, to minimize distractions, at a calculated speed based on each participant's step length (two-thirds of the leg length (m)) multiplied by a cadence constant of 1.33 (90 steps/60 seconds) [11,12]. Participants were asked to walk on a treadmill at a speed dictated by their stride length in a controlled environment. The flexion knee angle was measured by using two XSENS IMUs affixed to the participant's thigh and shank on their right leg and derived from the difference between the two angular positions in reference to a world coordinate system. After taking 250 steps to establish a baseline flexion knee angle (Baseline), each participant was asked to increase their walking flexion knee angle by 30 degrees and take 500 steps (Adaptation). The 30-degree increase was chosen to allow for a wider exploration of the task space. During this period participants were provided one of three different kinds of feedback depending on the group each participant were assigned to (Control, Reward, Punishment).

Visual feedback was displayed on a 60 cm screen, 85 cm in front of the participant, after every two steps with the right leg, completing a stride cycle, as shown in Fig 1A. Previous work has leveraged visual error feedback of lower extremity kinematics to alter locomotor patterns and induce learning [1,23,24]. To this end, we are utilizing the methods employed by these previous studies to compare with reinforcement feedback, to determine the effectiveness

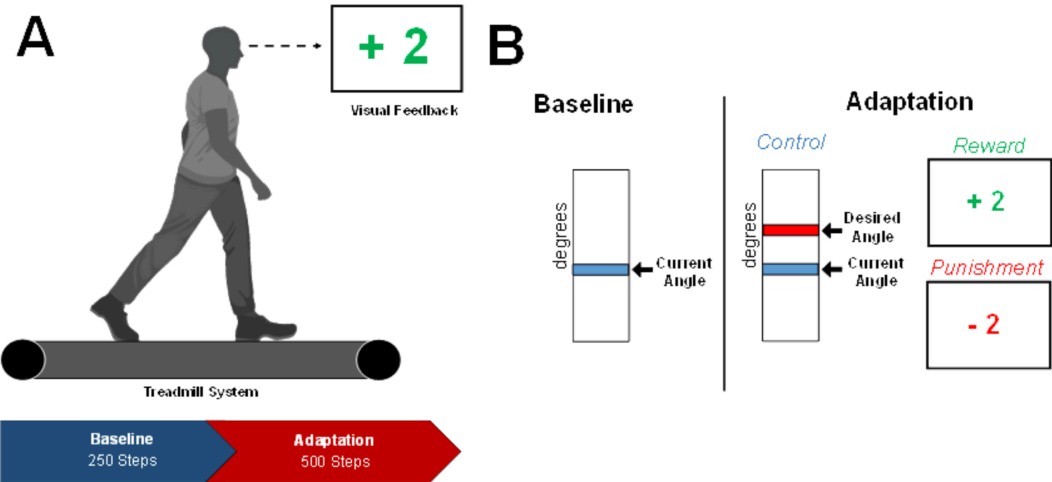

**Fig 1. A. Visual feedback presented on a screen, concurrently to walking.** Progression of task conditions. Duration was dictated by number steps. **B.** Example feedback based on group assignment from each task condition.

each type of feedback in altering locomotor adaptation and memory, which aligns with other studies in this domain [3,11,12]. As in Fig 1B, the Control group was provided with a vertical scale with their current and target line representing the desired maximum knee flexion angle. This target line remained in the same location throughout the experiment. The Reward and Punishment groups used a monetary scoring feedback system based on the difference between current and desired knee flexion angle. Participants were shown a number corresponding to a monetary gain or loss. The magnitude of scoring feedback was dependent on the amount of error in the previous stride cycle and followed these criteria:

- Reward: +4 points: meets desired angle; +3 points: within 10°; +2 points: within 20°; +1 point: within 30°; 0 points: exceeds or fewer than 30°.
- Punishment: 0 points: meets desired angle; -1 point: within 10°; -2 points: within 20°; -3 points: within 30°; -4 points: exceeds or fewer than 30°.

All groups started with a total of zero points. Those in the Reward group earned positive points, while those in the Punishment group accrued negative points. The Reward group was instructed that they begin with 0.00 USD and earn money based on their performance. The Punishment group was instructed that they began with 30.00 USD and lost money based on their performance. In addition, participants in the Control group were given the instructions of the reward or punishment groups, so as to control the effects of the script. To ensure equity in compensation, all participants were compensated, despite group assignments, with the full 30.00 USD at the end of the study, regardless of performance. Feedback presentation followed the same timing latency, ten milliseconds after completing a stride cycle. The task goal was to minimize error by matching the current knee flexion angle with the desired knee flexion angle. In this paper we chose to focus our attention on the first 500 steps; the 250 in the Baseline round and the first 250 of the Adaptation round, and tried to estimate the learning rate and retention factor of these time points. Further study may look at the additional time points.

To ensure homogeneity in the data collection, the groups were allocated in such a way as to have a similar number of males and females in each feedback group, and walked with

similar treadmill speeds. Further, in a preliminary data analysis for the homogeneity of participants in age, height and weight, one-way analyses of variance revealed there was no significant differences among the three feedback groups. For EHI, the $F$-statistic with 2 and 3 degrees of freedom was 2.857. Subsequently, $F = 0.116$ and $p$-value = 0.891 for Age, $F = 0.082$ and $p$-value = 0.921 for Height, $F = 0.320$ and $p$-value = 0.729 for Body Weight, and $F = 0.144$ and $p$-value = 0.866 for Step Length, respectively.

The data we collected can be found in the supporting information S4 File, the raw flexion angles of the participants walking; and S5 File, the flexion angles adjusted as the difference of from the participants' baseline angles.

## 3 Statistical methodologies

We consider the standard state space model for the knee flexion angle during walking with the retention factor and the learning rate as in [13]. To examine the effect of feedback, we separated the participants into groups by the different reward-punishment conditions. Let $y_{ijt}$ denote the knee angle in walking collected from the participant $i = 1, \ldots, n_j$ in the group $j = 1, \ldots, G$ at time $\tau_t$, $t = 0, 1, \ldots, T$. At each time, we establish a response model to characterize a group effect as the mean of knee angles of participants in the group with random noise. In terms of the progression of the knee angles of each participant over time, the angle of the next step is affected by a combination of the angle of the current step and the group effect shared at the current time. In other words, we may construct a state model to project the next knee angle based on the current knee angle for the retention portion and the group effect or the common behavior for the learning portion of the group. Let $\mathbb{A}_j(\tau_t)$ denote the retention factor and $\mathbb{B}_j(\tau_t)$ denote the learning rate from the participants in group $j$ at time $\tau_t$. The standard state space model can then be described by the response model and the state model such that, for $i = 1, \ldots, n_j$ and $j = 1, \ldots, G$,

$$
\begin{aligned}
\text{Response Model:} \quad y_{ijt} &= m_{jt} + \epsilon_{ijt}, \quad t = 0, \ldots, T, \\
\text{State Model:} \quad y_{ij,t+1} &= \mathbb{A}_j(\tau_t) y_{ijt} + \mathbb{B}_j(\tau_t) m_{jt}, \quad t = 0, \ldots, T-1,
\end{aligned}
\tag{1}
$$

where $m_{jt}$ is the mean knee angle at time $\tau_t$ across participants in group $j$ involving perturbation $\epsilon_{ijt}$.

In the model (1), we may utilize two approaches to estimate the mean angles $m_{jt}$. Using sequential analysis, the mean knee angles are time-varying parameters and a dynamic model for parameters can be estimated for each time. On the other hand, using a functional data analysis, the mean knee angles are considered as functional values at each time. Once the mean functions are estimated, their functional values evaluated at each time can be used. Based on the estimated mean angles, we may derive the time-varying retention factors and learning rates. The two approaches are described in the following subsections.

### 3.1 Sequential analysis with state space model

For the knee angle $y_{ijt}$ from the participant $i$ in group $j$ at time $\tau_t$, $i = 1, \ldots, n_j$, $j = 1, \ldots, G$ and $t = 1, \ldots, T$, we assume the perturbation $\epsilon_{ijt}$ to be Normally distributed with mean zero and variance $\sigma_{jt}^2$ for the response model in (1). Since the response is measured sequentially along with time $\tau_t$, the perturbations at two adjacent time points from the same participant, $\epsilon_{ijt}$ and $\epsilon_{ijt'}$ for $t \neq t'$, need not be statistically independent, while the perturbations from two participants at the same time, $\epsilon_{ijt}$ and $\epsilon_{i'j't}$ for $i \neq i'$ and $j \neq j'$, can be independent. To address the

dependency of $\epsilon_{ijt}$ and $\epsilon_{ijt'}$, we utilize a transition model for the variance of random perturbations for each group such that

$$\log\left(\sigma_{jt}^2\right) = a_j \log\left(\sigma_{j,t-1}^2\right) + u_{jt}, \quad t = 1,\ldots,T, \tag{2}$$

where $\sigma_{j,0}^2$ is an initial variance, $a_j$ is a transition coefficient and $u_{jt}$ is a Normal disturbance term with mean zero and fixed variance $\sigma_u^2$. Thus, we model the mean and variance of knee angles from the participants in each group over all time points with the state space model (1) and an additional autoregressive transition model (2) for state parameters $m_{jt}$ and $\sigma_{jt}^2$ respectively. In this paper, we set $a_j = 1$ and $\sigma_u^2 = 0.01$ and initialize the variance as $\sigma_{j,0}^2 = 1$.

Under the Bayesian framework, we assign a conjugate inverse Gamma prior distribution for the state parameter $\sigma_{jt}^2$ at the initial time point $\sigma_{j1}^2 \sim \text{Inv-Gamma}(A_{sj}, B_{sj})$, where we set $A_{sj} = 2$ and $B_{sj} = 1$ for this paper, and use the transition model specified in (2) after the initial time point instead of assigning a prior. For the other parameters, we allocate conventional priors such that $m_{jt} \sim \text{Normal}(A_{mjt}, B_{mjt}^2)$, for $j = 1,\ldots,G$ and $t = 1,\ldots,T$. In this paper, we set the hyperparameters as $A_{mjt} = 0$ and $B_{mjt} = 10$ to pursue vague priors.

We collect the Markov Chain Monte Carlo (MCMC) samples of parameters in the model for the estimation by using a Gibbs sampler [25] and dynamically weighted particle filter methods (DWPF) with dynamically weighted importance sampling [26]. Let $\boldsymbol{y}_t = (y_{11t},\ldots,y_{n_G Gt})$ denote a vector of responses for each participant at time $t$, $\theta_t = (m_{1t},\ldots,m_{Gt})$ denote a vector of model parameters and $\lambda_t = (\sigma_{1t}^2,\ldots,\sigma_{Gt}^2)$ denote a vector of state parameters that are varying by time. Further, let $\boldsymbol{y}_{1:t} = (\boldsymbol{y}_1,\ldots,\boldsymbol{y}_t)$ denote a sequence of responses from time $\tau_1$ to $\tau_t$, and $\lambda_{1:t} = (\lambda_1,\ldots,\lambda_t)$ denote the sequence of state parameters up to time $\tau_t$. Then, we use the full conditional distributions to generate MCMC samples for $\theta_t$ and use a dynamically weighted particle filter to estimate $\lambda_t$. The full conditional distributions are given by, for $j = 1,\ldots,G$,

$$m_{jt}|\cdot \overset{ind}{\sim} \text{Normal}(M_{jt}, V_{jt}), \quad t = 1,\ldots,T$$

$$\sigma_{j1}^2|\cdot \overset{ind}{\sim} \text{Inv-Gamma}(A_{sj}^*, B_{sj}^*),$$

where $V_{jt} = \left(\frac{n_j}{\sigma_{jt}^2} + \frac{1}{B_{mjt}^2}\right)^{-1}$, $M_{jt} = \left(\frac{\sum_{i=1}^{n_j} y_{ijt}}{\sigma_{jt}^2} + \frac{A_{mjt}}{B_{mjt}^2}\right) V_{jt}$, $A_{sj}^* = \frac{n_j}{2} + 2$ and $B_{sj}^* = \frac{1}{2}\sum_{i=1}^{n_j}(y_{ij1} - m_{j1})^2 + 1$. A full description of the dynamically weighted particle filter algorithm can be found in Supplementary File 1 as in [27].

Based on the population of particles, we may estimate the sequence of mean knee angles by taking the weighted average of particles at each time point. Because we have such a large weight-controlled population, on the order of thousands of particles, the weighted mean and the simple mean of particles are quite similar to each other. Thus for simplicity, in this paper, we use the simple mean of particles as the Bayesian estimate of parameters. Let $\hat{m}_{j1},\ldots,\hat{m}_{jT}$ denote the sequence of estimated mean knee angles. Further, let $\bar{y}_{jt} = \frac{1}{n_j}\sum_{i=1}^{n_j} y_{ijt}$ denote the observed mean knee angles for group $j = 1,\ldots,G$ at time $\tau_t$, $t = 1,\ldots,T$. Then, using the R package lavaan [28], we can estimate the retention factor $\mathbb{A}_j(\tau_t)$ and the learning rate $\mathbb{B}_j(\tau_t)$ by solving the following structural equation model such that

$$\bar{y}_{j,t+1} = \mathbb{A}_j(\tau_t)\bar{y}_{jt} + \mathbb{B}_j(\tau_t)\hat{m}_{jt}, \quad t = 1,\ldots,T-1. \tag{3}$$

All code used to run the dynamically weighted particle filter can be found in the supporting information S2 File.

## 3.2 Functional data analysis

As an alternative approach we consider a functional data analysis (FDA) by taking the sequentially measured knee angles $y_{ijt}$ from the participant $i = 1, \ldots, n_j$ in group $j = 1, \ldots, G$ over time $\tau_t$, $t = 1, \ldots, T$ as functional responses. In the state space model (1) the response model is expanded with a smooth function $f_j$ such that

$$
\begin{aligned}
y_{ijt} &= m_{jt} + \epsilon_{ijt}, \ i = 1, \ldots, n_j, \ j = 1, \ldots, G, \\
m_{jt} &= f_j(\tau_t) + \delta_{jt}, \ t = 1, \ldots, T,
\end{aligned}
\tag{4}
$$

where the mean knee angle $m_{jt}$ is treated as a random effect, $\delta_{jt}$ is the random discrepancy which explains the variation of group means, $\epsilon_{ijt}$ is the random error of individual participants and $\delta_{jt}$ and $\epsilon_{ijt}$ are mutually independent. The discrepancy and random error are assumed to follow the Normal distribution with zero means and constant variances $\sigma_{jd}^2, j = 1, \ldots, G$, and $\sigma_e^2$, respectively.

In this paper, specifically, we consider the cubic P-splines [29] as a smoothing function $f_j$. The $K$ knot points $\mu_1, \ldots, \mu_K$ are selected from time points $\tau_t$, $t = 1, \ldots, T$. The basis function $(\tau - \mu_k)_+^3$ has the value of $(\tau - \mu_k)^3$ if $\tau - \mu_k \geq 0$ and 0 otherwise, for knot $\mu_k$. The cubic P-splines can be expressed by

$$
f_j(\tau) = \beta_{j0} + \beta_{j1}\tau + \beta_{j2}\tau^2 + \beta_{j3}\tau^3 + \sum_{k=1}^{K} \beta_{j,k+3}(\tau - \mu_k)_+^3,
$$

where $\beta_{j0}, \ldots, \beta_{j,K+3}$ are regression coefficients. Specifically, we take $K = 19$ equally spaced knot points within the range of $0 \leq t \leq 500$ such that $\mu_1 = 25, \ldots, \mu_{19} = 475$.

As a Bayesian approach we assign the prior distributions for unknown parameters in the model (4). Let $\beta_j = (\beta_{j0}, \ldots, \beta_{j,K+3})^T$ denote a vector of regression coefficients for the mean of group $j$ and assign independent conjugate Normal prior distributions

$$
\beta_j \overset{iid}{\sim} \text{NORMAL}(\mathbf{0}, \alpha_j^{*-1}\mathbf{D}^-), \quad j = 1, \ldots, G,
$$

where $\alpha_j^* = \frac{\alpha_j}{\sigma_{jd}^2}$ is a smoothing parameter $\alpha_j$ scaled by the discrepancy $\sigma_{jd}^2$ and $\mathbf{D}^-$ is a generalized inverse matrix of diagonal matrix $\mathbf{D}$, which is a diagonal matrix with zeros for the first 4 diagonal elements and ones for the rest of diagonal elements. Regarding group-specific smoothing parameters and discrepancies, we assign independent and identical conjugate Gamma prior distributions and inverse Gamma prior distributions, $\alpha_j^* \overset{iid}{\sim} \text{GAMMA}(A_t, B_t)$ and $\sigma_{jd}^2 \overset{iid}{\sim} \text{INV-GAMMA}(A_d, B_d), j = 1, \ldots, G$, with hyperparameters $A_t$, $B_t$, $A_d$ and $B_d$, respectively. Likewise, for the variances of random error, we assign conjugate inverse Gamma distribution such that and $\sigma_e^2 \sim \text{INV-GAMMA}(A_e, B_e)$ with hyperparameters $A_e$ and $B_e$.

We estimate the model using Markov Chain Monte Carlo (MCMC) samples of parameters pulled from the joint posterior distribution. Utilizing a Gibbs sampler, we can collect the MCMC samples based on the full conditional distribution. Let $\mathbf{y}_{ij} = (y_{ij1}, \ldots, y_{ijT})^T$ denote the vector of responses, $\mathbf{m}_j = (m_{j1}, \ldots, m_{jT})^T$ denote the vector of mean angles and $\mathbf{X}$ denote the design matrix with the row vectors $\mathbf{x}_t = [1, \tau_t, \tau_t^2, \tau_t^3, (\tau_t - \mu_1)_+^3, \ldots, (\tau_t - \mu_K)_+^3]$ for $t = 1, \ldots, T$. Then, the full conditional distributions of parameters can be derived such that

$$\beta_j|\cdot \overset{ind}{\sim} \text{Normal}\left[(X^T X + \alpha_j D)^{-1} X^T m_j, (X^T X + \alpha_j D)^{-1}\sigma_{jd}^2\right],$$

$$m_j|\cdot \overset{ind}{\sim} \text{Normal}\left[\left(\frac{n_j}{\sigma_e^2} + \frac{1}{\sigma_{jd}^2}\right)^{-1}\left(\sum_{i=1}^{n_j} y_{ij} + X\beta_j\right), \left(\frac{n_j}{\sigma_e^2} + \frac{1}{\sigma_{jd}^2}\right)^{-1} I\right],$$

$$\alpha_j^*|\cdot \sim \text{Gamma}\left[A_t + \frac{K}{2}, B_t + \frac{1}{2}\left(\beta_j^T D\beta_j\right)\right],$$

$$\sigma_{j,d}^2|\cdot \overset{ind}{\sim} \text{Inv-Gamma}\left[A_d + \frac{T}{2}, B_d + \frac{1}{2}(m_j - X\beta_j)^T(m_j - X\beta_j)\right],$$

$$\sigma_e^2|\cdot \sim \text{Inv-Gamma}\left[A_e + \frac{\sum_{j=1}^G n_j + T}{2}, B_e + \frac{1}{2}\sum_{j=1}^G \sum_{i=1}^{n_j}(y_{ij} - m_j)^T(y_{ij} - m_j)\right],$$

for $j = 1, \ldots, G$. In this paper, we collect 1000 MCMC samples of parameters after 1000 burning iterations. From each set of MCMC samples of $\beta_j$, denoted $\beta_j^{(s)}$, we can derive the MCMC samples of the vector of functional responses such that $[f_j^{(s)}(\tau_1), \ldots, f_j^{(s)}(\tau_T)]^T = X\beta_j^{(s)}$, $s = 1, \ldots, S = 1000$. By taking the differences of functional responses we can also approximate the derivatives of functional responses at time $\tau_t$ such that $f_j'(\tau_t) = f_j^{(s)}(\tau_{t+1}) - f_j^{(s)}(\tau_t)$, $t = 1, \ldots, T-1$. The Bayesian estimates of mean function and its derivative at time $t$ can be obtained by taking the averages of MCMC samples, that is $\hat{f}_j(\tau_t) = \frac{1}{S}\sum_{s=1}^S f_j^{(s)}(\tau_t)$ and $\hat{f}_j'(\tau_t) = \frac{1}{S}\sum_{s=1}^S f_j'^{(s)}(\tau_t)$. The credible intervals of the mean function and its derivative can be estimated by taking the quantile values of $f_j^{(s)}(\tau_t)$ and $f_j'^{(s)}(\tau_t)$, respectively. It should be noted that the credible intervals of derivatives of functions are generally not easy to calculate. However, in this paper, we can readily use the quantiles of MCMC samples in the calculation of credible intervals under a Bayesian framework.

Regarding the estimation of the retention factors $\mathbb{A}_j(\tau_t)$ and the learning rates $\mathbb{B}_j(\tau_t)$, the FDA provides time-varying functions of the retention factors and the learning rates. From the state space model (1), the group mean function $m_{jt}$ is estimated by $\hat{f}_j(\tau_t)$ in FDA. Regarding the state model as a regression model for the knee angles at time $t+1$ on the knee angles at time $t$ with disturbance term $\mathbb{B}_j(\tau_t)m_{jt}$, the retention faction $\mathbb{A}_j(\tau_t)$ is a slope that represents the change rate of knee angles from current time to the next. Hence, for the estimated mean functions, the retention factor $\mathbb{A}_j(\tau_t)$ or the change of responses by the time is estimated by $\hat{f}_j'(\tau_t)$ in FDA. Hence, denoting $\mathbb{A}_j(\tau_t)$ and $\mathbb{B}_j(\tau_t)$ as the function of retention factors and the function of learning rates respectively, we apply the results from FDA to the transition model in (1) and estimate the functions with the observed mean knee angles $\bar{y}_{jt}$ such that

$$\mathbb{A}_j(\tau_t) = \hat{f}_j'(\tau_t),$$
$$\mathbb{B}_j(\tau_t) = \frac{\bar{y}_{j,t+1} - \hat{f}_j'(\tau_t)}{\hat{f}_j(\tau_t)}. \tag{5}$$

The credible region of $\mathbb{A}_j(\tau_t)$ is equivalent to that of $\hat{f}_j'(\tau_t)$ and the credible region of $\mathbb{B}_j(\tau_t)$ can be calculated by taking the quantile values of $\frac{\bar{y}_{j,t+1} - \hat{f}_j'^{(s)}(\tau_t)}{\hat{f}_j^{(s)}(\tau_t)}$, $s = 1, \ldots, S$. It should be noted that $\mathbb{A}_j(\tau_t)$ in this instance actually shows the change of the retention factor over time, and it can be interpreted as $\mathbb{A}_j(\tau_t) - 1$ at each time $\tau_t$ to compare with the results from the state space model in Sect 3.1.

All code used to run the functional data analysis can be found in the supporting information S3 File.

## 4 Results

### 4.1 Sequential analysis with state space model

The dynamically weighted particle filter (DWPF) estimated the group means and variances in the response model at (1) with 95% credible intervals as shown in Fig 2. The estimated mean angles have shown consistently below the observed mean angles. However, it is not surprising because the observed knee angles from the same participant are not independent and their sample means are well known to be biased. The estimated means have shown a jump when the time is greater than 250 in all three groups. The estimated variances have stayed at a low level with narrow credible interval when the time is less than 250. After that they increased with wider credible intervals and have become stabilized with reduced width of credible intervals in all groups. Fig 3 shows the three groups laid over each other to show where they deviate from one another.

Using the estimated mean functions $\hat{m}_{jt}$ in the Sect 3.1, for $j = 1, \dots, 3$ and $t = 1, \dots, 500$, and the observed mean angles we can estimate the retention factor $\mathbb{A}_j(\tau_t)$ and the learning rate

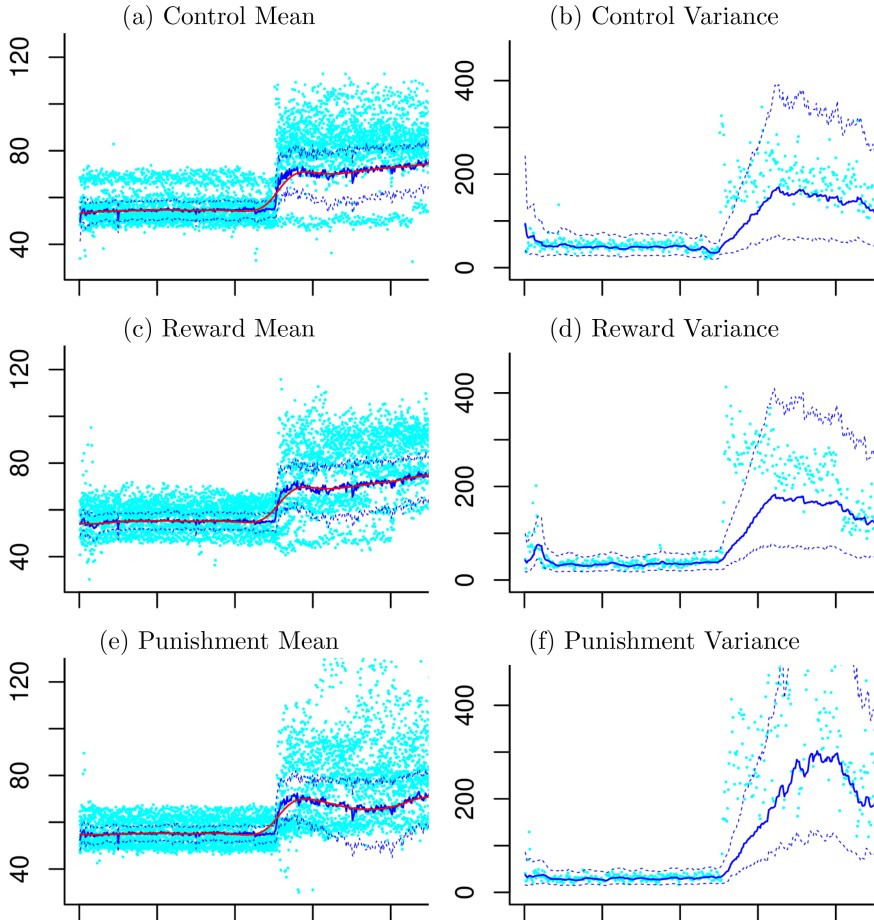

**Fig 2. Estimated Means and Variances.** In (a), (b) and (c) light blue dots are observed angles, blue dotted lines are 95% credible regions, blue solid lines are the estimated mean functions and red lines are the smoothed mean function, respectively. In (d), (e) and (f) light blue dots are observed mean angles, blue dotted lines are 95% credible regions, blue solid lines are the estimated variance functions.

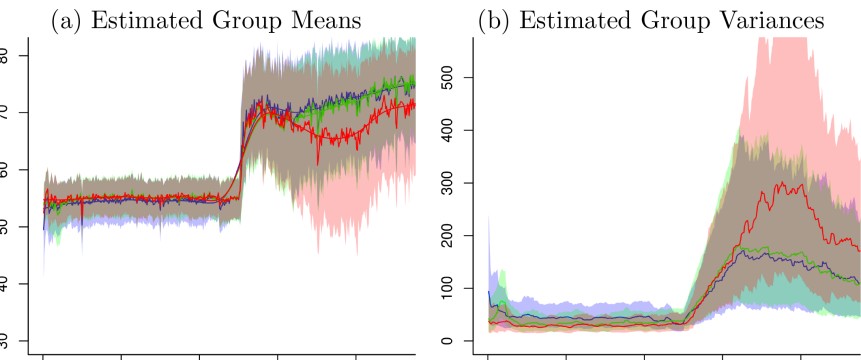

**Fig 3. Estimated Means and Variances**. In (a), the estimated means, the smoothed mean functions and the 95% credible regions are shown for each group. In (b), the estimated variance functions and the 95% credible regions are shown for each group. Blue lines denote the control group, green lines the punishment group and red lines the reward group.

$\mathbb{B}_j(\tau_t)$ for each group. Table 1 presents the estimated $\mathbb{A}_j(\tau_t)$'s and $\mathbb{B}_j(\tau_t)$'s and the results of their chi-squared difference tests to examine if three groups produce different $\mathbb{A}_j(\tau_t)$'s and $\mathbb{B}_j(\tau_t)$'s. Structural equation modeling was used to estimate the parameters. It has been shown that chi-squared differences between nested SEM models are equivalent to the standard analysis of variance (ANOVA) tests in [30]. The retention factors $\mathbb{A}_j(\tau_t)$'s have been estimated to be close to 1 in all groups with a statistically significant difference; whereas the learning rates $\mathbb{B}_j(\tau_t)$'s have been estimated to around 4% in the Control group, 5% in the Punishment group and 12% in the Reward group, a statistically significant difference.

Using the individual sequence of the observed knee angles instead of the group mean angles, we have also estimated $\mathbb{A}_j(\tau_t)$ and $\mathbb{B}_j(\tau_t)$ for each individual participant. The distributions of the individual estimated $\mathbb{A}_j(\tau_t)$'s and $\mathbb{B}_j(\tau_t)$'s within group are shown in the boxplots in Fig 4. Interestingly, the distribution of estimated $\mathbb{A}_j(\tau_t)$' in the control group have shown significantly lower levels than those in the punishment group and the reward group; whereas the distributions of the individual estimated $\mathbb{B}_j(\tau_t)$'s of three group show similar levels. This seems to contradict the results in Table 1, which shows the results for the group learning and retention rates. However, that can be explained that some of participants especially in the control group have quite unusual patterns of knee angles compared to the other participants.

**Table 1. Estimates and Inferences for Retention Factor $\mathbb{A}$ and Learning Rate $\mathbb{B}$ for Model using Particle Filter Means.**

| | Group | Estimate | Std.Error | Model | $\chi^2$ | Difference | $p$-value |
|---|---|---|---|---|---|---|---|
| $\mathbb{A}$ | Control | 0.9404 | 0.0091 | Null | 4714.3 | 64.968 | <0.0001 |
| | Punishment | 0.9394 | 0.0045 | Full | 4649.4 | | |
| | Reward | 0.8889 | 0.0048 | | | | |
| $\mathbb{B}$ | Control | 0.0477 | 0.0073 | Null | 4761.4 | 112.09 | <0.0001 |
| | Punishment | 0.0567 | 0.0042 | Full | 4649.4 | | |
| | Reward | 0.1231 | 0.0054 | | | | |

Note: The left three columns display the estimated $\mathbb{A}_j(\tau_t)$ and $\mathbb{B}_j(\tau_t)$ and the right four columns present the difference of $\chi^2$ tests, which are equivalent to the standard analysis of variance tests. The full model considers the effects of the treatments separately, and the null model considers them to be equivalent.

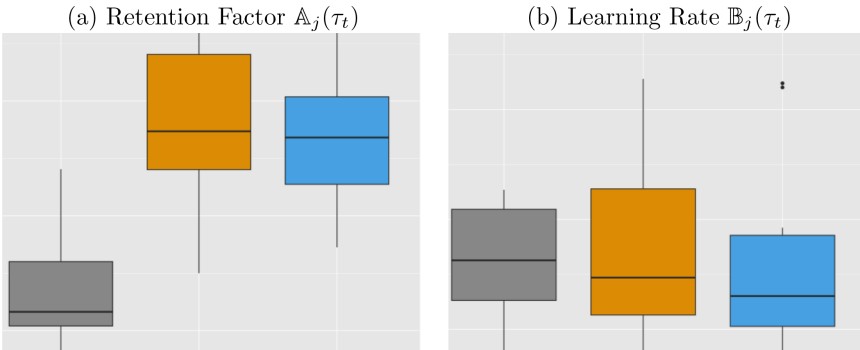

(a) Retention Factor $\mathbb{A}_j(\tau_t)$          (b) Learning Rate $\mathbb{B}_j(\tau_t)$

**Fig 4. Estimates for the Retention Factor and Learning Rate for the Three Treatments.** Boxplots are based on the estimated values for each case.

To evaluate the performance of the particle filter, we considered a simpler model using a structural equation model without the particle filter mean at time $t$ in the term containing $\mathbb{B}$, and replacing it with the sample mean at time $t$. The results of the models are given in Table 2. This simpler model finds very similar results to the model containing the particle filter mean, slightly over-estimating $\mathbb{A}$ and slightly under-estimating $\mathbb{B}$ compared with the model in Table 1. Both models find that the coefficients $\mathbb{A}$ and $\mathbb{B}$ are significantly different between the treatments. This suggests that the particle filter does efficiently and effectively model the mean of the data.

## 4.2 Functional data analysis

The flexion knee angles data used in the previous subsection have been fitted by the Bayesian cubic P-splines. The estimated group mean functions, $\hat{f}_j(\tau_t)$ in the Sect 3.2, for three groups are presented in Fig 5. It should be noted that $\hat{f}_j(\tau_t)$ is a smooth version of $\hat{m}_{jt}$ in the Sect 3.1. In all three groups, the mean angles have been increased in the average with a jump at time 250, while the control has a stabilized trend, the punishment has an increasing trend and the reward has a slightly decreasing trend after the jump.

Using the estimated mean functions $\hat{f}_j(\tau_t)$ and the related derivatives $\hat{f}'_j(\tau_t)$ in Sect 3.2, the time-varying retention factors $\mathbb{A}_j(\tau_t)$ and the learning rates $\mathbb{B}_j(\tau_t)$ have been obtained as shown in Fig 6. While the three group take similar shapes in the function $\mathbb{A}_j(\tau_t)$ and $\mathbb{B}_j(\tau_t)$,

**Table 2. Estimates and Inferences for Retention Factor $\mathbb{A}$ and Learning Rate $\mathbb{B}$ for Model without Particle Filter Means.**

|  | Group | Estimate | Std.Error | Model | $\chi^2$ | Difference | $p$-value |
|---|---|---|---|---|---|---|---|
| $\mathbb{A}$ | Control | 0.9729 | 0.0217 | Full | 4980.7 | 29.649 | <0.0001 |
|  | Punishment | 0.9563 | 0.0039 | Null | 4951.1 |  |  |
|  | Reward | 0.9371 | 0.0036 |  |  |  |  |
| $\mathbb{B}$ | Control | 0.0217 | 0.0049 | Full | 5018.0 | 66.899 | <0.0001 |
|  | Punishment | 0.0431 | 0.0038 | Null | 4951.1 |  |  |
|  | Reward | 0.0746 | 0.0044 |  |  |  |  |

Note: The left three columns display the estimated $\mathbb{A}_j(\tau_t)$ and $\mathbb{B}_j(\tau_t)$ and the right four columns present the difference of $\chi^2$ tests, which are equivalent to the standard analysis of variance tests. The full model considers the effects of the treatments separately, and the null model considers them to be equivalent.

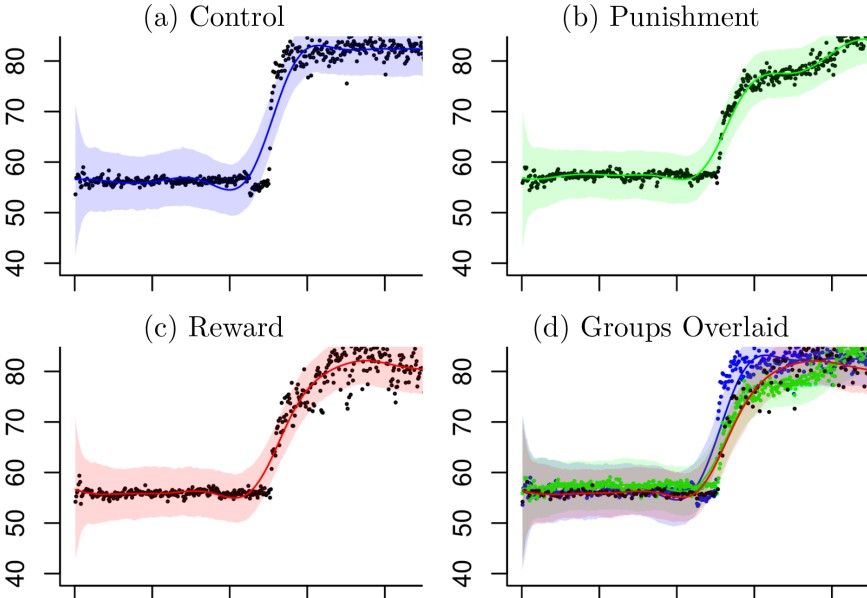

**Fig 5. Estimated Mean Functions.** The dots are observed means of knee angles, the lines are fitted curves and the shaded areas are 95% credible intervals.

the jump of knee angles at time 250 have brought a peak in the retention factors $\mathbb{A}_j(\tau_t)$. The learning rates $\mathbb{B}_j(\tau_t)$ take a drop just before time 250 and a jump afterward in all groups. Regarding the 95% credible intervals the estimated retention factor functions $\mathbb{A}_j(\tau_t)$ have wider intervals than the learning rate functions $\mathbb{B}_j(\tau_t)$ have. While $\mathbb{A}_j(\tau_t)$ take similar width except for the both ends, $\mathbb{B}_j(\tau_t)$ take an increasing width as the time increased.

Based on the 95% credible intervals the difference of $\mathbb{A}_j(\tau_t)$ and $\mathbb{B}_j(\tau_t)$ can be examined. As shown in Fig 7, comparing three groups from Fig 6 against each other, the credible intervals of $\mathbb{A}_j(\tau_t)$ overlap in all time, indicating no significant difference between groups. The credible intervals of $\mathbb{B}_j(\tau_t)$ shows separated intervals in the beginning and after time 250, which indicates significantly different effects of the groups.

## 5 Conclusion

We have utilized two approaches to examine the retention factor and the learning rate in the changes of knee angles during striding under three different feedback conditions, the state space model and the functional data analysis. Both approaches have led to compatible results, while the method based on the functional data analysis has allowed more flexible inference.

In the state space model approach, the retention factor $\mathbb{A}_j(\tau_t)$ that shows the relationship between two consecutive observations has been turned out about $\mathbb{A}_j(\tau_t) = 0.97$ without significant difference by the types of feedback. That is, the knee angles in the following steps have been about 97% of the knee angles of preceding step in the average no matter what type of feedback has been given to the participant. Meanwhile, in the functional data analysis approach, the retention factor has been varying while it mostly remains zero, $\mathbb{A}_j(\tau_t) \approx 0$, and has a peak in the middle at time 250. It again has shown that the angles of following steps have been similar to the angles in the preceding steps except for the steps around time 250, when the phase shifts from Baseline to Adaptation, without any significantly different effect of the feedback.

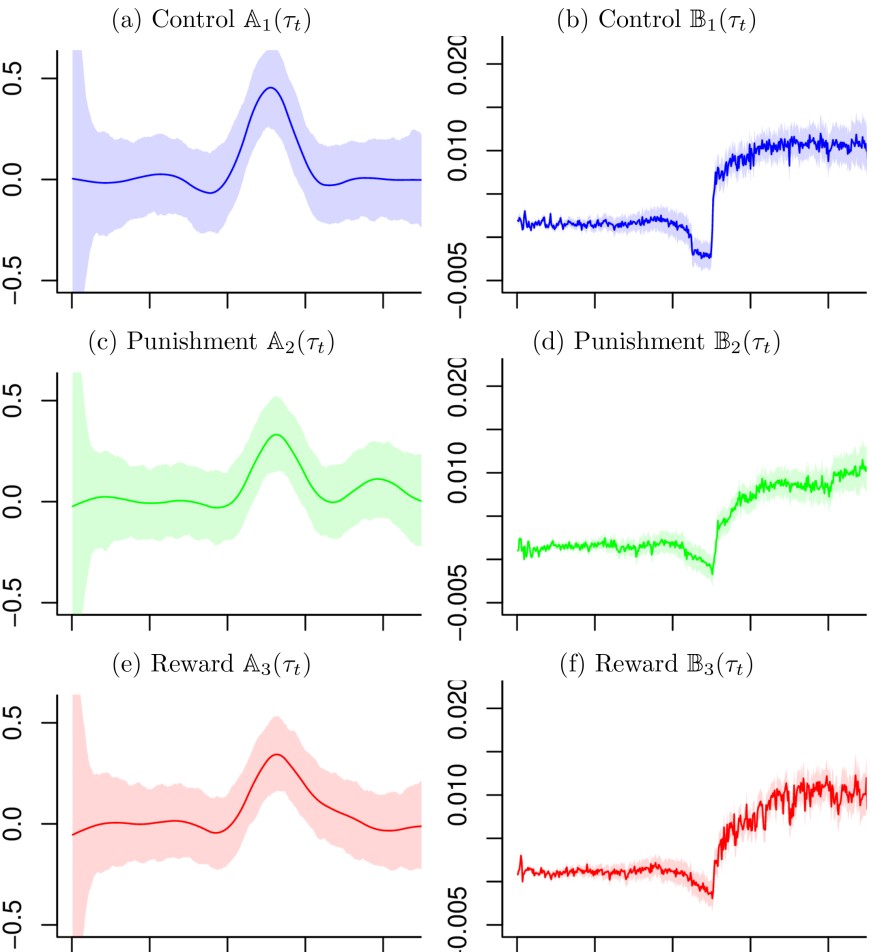

**Fig 6. Estimated Retention Factor** $\mathbb{A}_j(\tau_t)$ **and Learning Rate** $\mathbb{B}_j(\tau_t)$**.** The lines are estimated $\mathbb{A}_j(\tau_t)$ and $\mathbb{B}_j(\tau_t)$, and the shade areas are 95% credible intervals.

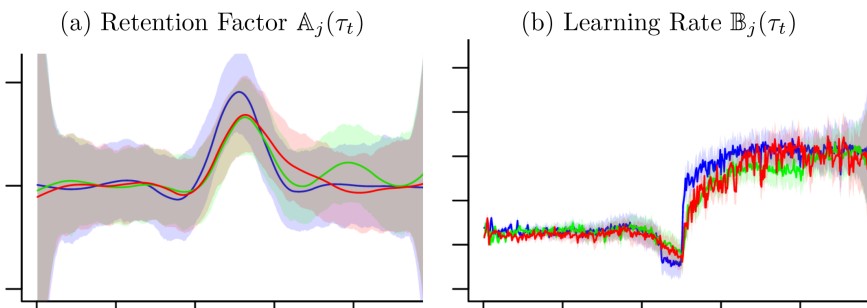

**Fig 7. Estimates for the Retention Factor and Learning Rate.** Blue lines and light blue area are estimated functions and 95% credible regions for the control group, green lines and light green area are those for the punishment group and red lines and light red areas are those for the reward group, respectively.

Regarding the learning rate $\mathbb{B}_j(\tau_t)$ that indicates the contribution of projected knee angle in the preceding step, the state space model approach has shown less than 5%. However, different rewards and punishments have resulted in significantly different contributions. Similarly, the functional data analysis approach has also indicated that the projected angles of the preceding step has little contribution to the angle of the next step and the effect of feedback has been significant. This implies that the use of reinforcement methods, reward-based or punishment-based, does seem to have an impact on participants learning to increase the flexion angles of their knee while walking. In addition, the contribution has been substantially increased after the peak of retention rate $\mathbb{A}_j(\tau_t)$.

Past work in the field concerning learning and retention rates [9,13,15] have dealt with short timescales and have relied on simpler methods of analysis, namely ANOVA. These methods are limited in properly considering large datasets collected over larger timescales and can require large computational loads.

The functional data analysis method allows for elegant analysis of the changing values of the retention factor and the learning rate over time. Also, by overlaying the estimates and the credible intervals makes inference simple and easily interpretable, by looking for the areas where the credible intervals do not overlap. However, one drawback of FDA is that if you want to update after adding new data, everything needs to be rerun on the full set of the data, both the original data and the new additions. The dynamically weighted particle filter approach lets us consider only the new data point by carrying the particles forward based on the results of the prior analysis, meaning we can quickly and efficiently incorporate new data.

The models used in this paper provide more efficient and flexible methods for estimating the parameters and the confidence intervals. The confidence intervals in Figs 6 and 7 are examples that can be difficult to estimate cleanly, but these models offer easy solutions for finding credible intervals. Applying these methods to further data sets will provide an opportunity to further develop the methods.

## Supporting information

**S1 File. Dynamically Weighted Particle Filter Model Algorithm.pdf** A full description of the dynamically weighted particle filter algorithm.
(PDF)

**S2 File. Full DWPF Code.R** All code used to run the dynamically weighted particle filter and use those particles to estimate the parameters of the state space model.
(R)

**S3 File. Full FDA Code.R** All code used to run the functional data analysis and estimate the means, the learning rate and retention factor and credible intervals for each.
(R)

**S4 File. Adaptation Peak Data.csv** Dataset of flexion knee angles, of which we considered the first 500 time points.
(CSV)

**S5 File. Adaptation Peak Data Baseline.csv** Dataset of baseline flexion knee angles for each participant.
(CSV)

## Author contributions

**Data curation:** Christopher M. Hill.

**Formal analysis:** Torin Quinlivan, Kacy Kane, Duchwan Ryu.

**Funding acquisition:** Christopher M. Hill.

**Investigation:** Torin Quinlivan, Kacy Kane, Christopher M. Hill, Duchwan Ryu.

**Methodology:** Torin Quinlivan, Kacy Kane, Duchwan Ryu.

**Writing – original draft:** Torin Quinlivan, Duchwan Ryu.

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
