## [Decision Letter · Decision Letter 0]

17 Apr 2025

PONE-D-24-38528Comparison of sequential data analysis and functional data analysis for locomotor adaptationPLOS ONE

Dear Dr. Quinlivan,

Thank you for submitting your manuscript to PLOS ONE. After careful consideration, we feel that it has merit but does not fully meet PLOS ONE’s publication criteria as it currently stands. Therefore, we invite you to submit a revised version of the manuscript that addresses the points raised during the review process.

The manuscript does not meet PLOS ONE’s standards for publication. Major revisions are necessary to clearly define the research question and ensure fair comparisons between groups, to clarify the statistical modeling and indexing, to control for confounding variables, to improve the clarity of figures, descriptions, and language throughout the manuscript.

After addressing these substantial issues detailed by the reviewers, the manuscript may be suitable for resubmission and reevaluation.

Major

The research question is not clearly defined, and the contribution is not articulated in relation to existing literature.

A critical experimental design flaw undermines the comparison between groups: the sensory group had access to both current and target knee angle feedback, giving them a significant advantage over the other groups. This invalidates direct comparisons and weakens the interpretation of feedback effects.

The statistical modeling is inconsistently presented and confusing. Notation is unclear, code-derived variable names are mixed with formal equations, and indices (e.g., subject ID) are used inconsistently.

Equations (especially Eq. 1 and Eq. 6) are not properly justified or explained, and there is confusion between time-series data and step-based data, calling into question the validity of the model implementation.

Important covariates (e.g., age, sex, height, weight) were not controlled for across groups, introducing potential bias.

Figures and captions are unclear or incomplete, making it difficult to interpret key results.

The task design lacks balance across experimental groups due to unequal feedback access.

The methods section contains redundancies, and task details are insufficiently illustrated—reviewers suggest including figures or videos to clarify the setup.

The control group is not clearly described, and figure labeling (especially in Figs. 1, 3, 4) is poor.

The manuscript contains grammatical issues and typos.

The abstract and introduction are too vague and do not frame the problem or findings clearly.

The mathematical and algorithmic descriptions require major clarification.

We look forward to receiving your revised manuscript.

Kind regards,

Gennady S. Cymbalyuk, Ph.D.

Academic Editor

PLOS ONE

Journal Requirements:

Reviewers' comments:

Reviewer's Responses to Questions

**Comments to the Author**

1. Is the manuscript technically sound, and do the data support the conclusions?

Reviewer #1: No

Reviewer #2: Partly

2. Has the statistical analysis been performed appropriately and rigorously? 

Reviewer #1: No

Reviewer #2: No

3. Have the authors made all data underlying the findings in their manuscript fully available?

Reviewer #1: Yes

Reviewer #2: Yes

4. Is the manuscript presented in an intelligible fashion and written in standard English?

Reviewer #1: Yes

Reviewer #2: Yes

5. Review Comments to the Author

Reviewer #1: This paper is studying how learning rates depend on sensory, reward, and punishment feedback. Authors consider state space models for dynamically changed learning rates as well as functional data analysis for the learning rates and the effect of the feedback.

However, sensory groups have advantages because they can see current and target lines of peak knee flexion angle, while reward and punishment groups can only roughly estimate deviation of target without knowing whether subject needs to more or less flex knee angle.

Also, state model description is questionable. Formula (1) shows ‘mean knee angle’ for particular time and group as sum of knee angle and perturbation for particular time and group and subject. There is no index ‘I’ in left part of equation, but there is one in right part of equation. One may consider that the right part of equation was summarized across all subjects, another one may suggest different operations on omitted index.

As well ‘knee angle’ used in right part of equation denote the knee angle at time t = 0,1,...,T in walking. While early in methods there is information about 250 steps for baseline and 500 steps for adaptation.

In provided code authors used 500 samples for analysis, assuming that knee angle has one value for each step, not for each time instance as in the model description.

Figures (1,3,4) are hard to see differences, maybe it would be more clear if angles of all groups were plotted on one panel with different colors, similar to figure(5).

These model descriptions problems could be fixed but comparison of sensory group with other two is not correct because sensory groups have advantages mentioned earlier.

Reviewer #2: The authors study an adaptation task for walking patterns using two different statistical approaches. Overall, the manuscript is very shallow: the literature review in the introduction is very brief, and both the open problem and how the authors approach it is not clearly stated. The experimental description is fine and understandable, but the relevance of the experiment is not stated in the intro nor anywhere else in the paper. The statistical model description is obscure to say the least and notation is not consistent, bringing serious concerns about the study (see below) -- e.g., equations are not interpreted, and are only given mathematically, and it seems that the R code notation is being naively translated into the mathematical description and vice-versa, raising serious confusion. There is no discussion to compare the current findings with previous works in the literature, and the conclusions are not (or do not seem to be) articulated with the results displayed in the figures. The group division is arbitrary (random), and the authors do not control for some parameters that influence the findings (e.g., body height and weight, age or sex), raising serious concerns about the statistical significance of the findings.

Considering the PLoS ONE criteria, this work is not acceptable for publication in its current form. A thorough revision must be done, after which the paper could be resubmitted and reevaluated.

# Major concerns

1. The stated parameters for the sampled population (age, sex, body height and weight) should have been done after the division into three groups (Reward, Punishment, Sensory), not prior. In this way you would have been able to test the influence of these parameters on the measured difference between groups. With only the "global" average of these parameters, you cannot safely test whether height, for example, or age, influence the knee angle for a given condition. Ideally, to minimize the effects of these parameters, you'd want each of the groups (Reward, Punishment, Sensory) to have matching mean age, height, weight and M-F sex ratio. Did you take this into account? If you didn't, did you consider the possibility that these variables could be statistically influencing your results? Otherwise, what is your expectation about the effect these parameters have on your task, and why?

2. Eq 1: is there an implied summation over i (participants) in the m definition? Please, use an explicit notation. Also, define A and B. Are these tensors? What rank? Be more explicit and consistent with your notation. Justify Eq. 1, why would the knee angle y be required to go to the mean m? Wouldn't the learning be successful if the participant achieved the required flexing angle (30°) instead of the mean m? What is A and B from the physical point of view? What is the meaning of "retention" and "learning rate"? What kind of learning is going on? How is this learning reflected in the locomotor-related brain systems? Clarify it in the manuscript. When you say "we solve the transition model" (line 139), be explicit: what is this? Inverting Eq. 1 for each time step in order to determine A and B? Also clarify it in the manuscript. What do you use A and B for? Are they displayed in some figure? What are the quantities of interest, and how are they related to A and B? Do you take temporal averages of A and B? Although from the results, these A and B seem to be just numbers... The presentation of the model is very confusing, and must be thoroughly expanded and clarified (see also the next two comments).

3. Line 162: what's "IG"? Wasn't G the number of groups (i.e., G=3)? G also appears in between lines 280-281. The authors should be more specific when describing their model. If one want the main manuscript to be more concise, then a supplementary file contaning a detailed specific description should be provided. Are A_sj and B_sj (line 162-163) elements of the A and B tensors? But shouldn't A be rank 3 and B rank 2? The authors also write A_mjt, B_mjt (lines 165-166). Are these "mjt" or "sj" the element indices? So are these tensors rank 2 or 3? The model description is very obscure and sloppy and not precise at all, again lacking consistency in the notation within the paper. I noticed that the authors developed their own code in R. However, if the regression method is THE MAIN novelty of this paper, this must be clearly stated in the abstract and introduction. Otherwise, if the regression for each model was simply an implementation of a previously published regression method (which seems to be the case), then there's no need to explicitly state the algorithm (e.g., lines 226-234; or eqs between lines 280-281), provided that a clear mention and reference to the original work is given. I advise the algorithm to be moved to a Supplementary File, and focus Section 3 in a precise and rigorous description of the models (something that is not done yet). Mathematical details must be thoroughly improved, clearly stating what each symbol means, and what is the role of subscripts (elements in tensors or just variable names? From code inspection, they look like just variable names, so everything is very confusing).

4. If these tensors have independent elements "A_mjt" or "B_st", then how can they be equated to a function of time with a single index for group j (eq 6)? I strongly advise a full revision of the model sections (sec 3.1 and 3.2), including a preamble in Section 3 where the authors write an Eq. 1 that generalizes both approaches (SDA and FDA), and then in each subsection, the description is then reduced to that particular case. Again, stating the details of the algorithm that was used to fit the A and B tensors is not important if it was done using a software package, and can be skipped for clarity along with citing the approapriate package. DO NOT MIX code notation with the mathematical description of the code for the sake of reproducibility and clarity. The details of the algorithm must only be stated (separately in a Supporting file) if this was a major development of you paper (which I assume it wasn't, otherwise it isn't clear and it must be highlighted in the introduction that your contribution is a new algorithm for fitting statistical models).

5. Results: what is the "Control" group? It was not described in Methods. Is the "Sensory" the "Control" group? This must be clarified. Fig 2 has unreadable labels. Figures and Tables' captions are not informative nor descriptive enough. Lines 337-338, the authors say they use a simpler model to evaluate the performance of particle filter. This MUST BE clearly described in Methods (Section 3).

6. Conclusion: "Both approaches have led to compatible results, while the method based on the functional data analysis has allowed more flexible inference.". This is not clear in the manuscript. The concluding remarks are too shallow: how do your results compare to previously found results in the literature? Did you just confirm previous findings? What is your contribution? What advantages/setbacks, compared to previous works, came from your statistical methods? I do not see how the figures you showed can help answering these questions, so include a Discussion articulating the relation between the figures and your findings.

# Other concerns that must be addressed

7. A figure or video illustrating the task and the different condtions (Reward, Punishment, Sensory) is highly advisable (even though it is written in lines 90-99). For example, what was the visual feedback? What are the walking patterns (authors state there are five distinct patterns, line 74). How could the participant know they are flexing their knees by an extra 30° (line 86)? What were the task direction read to the participants? These directions can be attached to the submission, and a Figure can be added to the main manuscript, and a video could be appended as well.

8. The abstract is not clear. It is too concise and does not frame the work into the big picture (e.g., what is the problem being addressed, or why is it important).

It does not clearly state what the authors did to address the problem either. It only states briefly that two models were compared, but what did the authors find? What is the motivation for both of these models in relation to the open problem?

9. Intro, lines 23-24: "The effect of reinforcement on skill learning are complex, multifaceted, and in need of a consensus.". It is not clear in the introduction why it is in need of "consensus". Please, clarify it in the manuscript.

10. There are many typos (e.g., line 9: "there are two TYPE of errors"; line 20: "affects" should be "effects"; line 29: "providing" -> "provided"; etc).

Revise the whole text for similar problems.

11. The authors mention "motor adaptation paradigm" (line 34 Intro), but do not detail the task. Give a quick outline of the task in the introduction before talking about "knee angles" (line 37).

12. "The derivative of fitted curves can be used for the estimation of the retention rates". Why? Either cite a work that's done that, or quickly explain it if it hasn't been done before.

13. Sensor positioning and data acquisition (lines 112-114) was already stated previously (lines 81-84).

14. "only the participant IDs distinguished the entries". This is not clear: was the actual "ID" (official document of the participant) used? (probably not). So rephrase... "a number was assigned to each participant" etc.

15. Line 148: "model in (1)". Unclear: Eq. (1) or Ref (1)?

16. Fig. 1, bottom row (Punishment) has mean and variance panels switched.

6. PLOS authors have the option to publish the peer review history of their article (what does this mean?). If published, this will include your full peer review and any attached files.

Reviewer #1: No

Reviewer #2: No

---

## [Author Response · Author response to Decision Letter 1]

17 Jun 2025

We have attached a letter with replies to all the comments the reviewers and editors made, please refer to that.

---

## [Decision Letter · Decision Letter 1]

22 Jul 2025

PONE-D-24-38528R1Comparison of sequential data analysis and functional data analysis for locomotor adaptationPLOS ONE

Dear Dr. Quinlivan,

Thank you for submitting your manuscript to PLOS ONE. After careful consideration, we feel that it has merit but does not fully meet PLOS ONE’s publication criteria as it currently stands. Therefore, we invite you to submit a revised version of the manuscript that addresses the points raised during the review process.

Please, make the requested corrections:Minor corrections to clarify model description:

Pp.170 Since retention factor A and learning rate B do not have index (i), participants (plural) should be used instead of participant.

pp.173 It should be ‘mean knee angle across participants’ instead of ‘for the participants’.

We look forward to receiving your revised manuscript.

Kind regards,

Gennady S. Cymbalyuk, Ph.D.

Academic Editor

PLOS ONE

Journal Requirements:

Reviewers' comments:

Reviewer's Responses to Questions

**Comments to the Author**

1. If the authors have adequately addressed your comments raised in a previous round of review and you feel that this manuscript is now acceptable for publication, you may indicate that here to bypass the “Comments to the Author” section, enter your conflict of interest statement in the “Confidential to Editor” section, and submit your "Accept" recommendation.

Reviewer #1: All comments have been addressed

Reviewer #2: All comments have been addressed

2. Is the manuscript technically sound, and do the data support the conclusions?

Reviewer #1: Yes

Reviewer #2: Yes

3. Has the statistical analysis been performed appropriately and rigorously? 

Reviewer #1: Yes

Reviewer #2: Yes

4. Have the authors made all data underlying the findings in their manuscript fully available?

Reviewer #1: Yes

Reviewer #2: Yes

5. Is the manuscript presented in an intelligible fashion and written in standard English?

Reviewer #1: Yes

Reviewer #2: Yes

6. Review Comments to the Author

Reviewer #1: Minor corrections to clarify model description:

Pp.170 Since retention factor A and learning rate B do not have index (i), participants (plural) should be used instead of participant.

pp.173 It should be ‘mean knee angle across participants’ instead of ‘for the participants’.

Reviewer #2: All my concerns were addressed by the authors. The manuscript was significantly improved and is now suitable for publication.

7. PLOS authors have the option to publish the peer review history of their article (what does this mean?). If published, this will include your full peer review and any attached files.

Reviewer #1: No

Reviewer #2: **Yes: **Mauricio Girardi-Schappo

---

## [Author Response · Author response to Decision Letter 2]

22 Jul 2025

We deeply appreciate your careful review and helpful comments on the manuscript. In

what follows we describe how we have addressed the review comments to the authors:

Reviewer #1

◦ pp.170 Since retention factor A and learning rate B do not have index (i), partic-

ipants (plural) should be used instead of participant.

Thank you for catching that. The manuscript has been updated to “Let Aj (τt) de-

note the retention factor and Bj (τt) denote the learning rate from the participants

in group j at time τt. ”.

◦ pp.173 It should be ‘mean knee angle across participants’ instead of ‘for the par-

ticipants’.

Thank you for catching that. The manuscript has been updated to “where mjt is

the mean knee angle at time τt across participants in group j involving perturba-

tion ϵijt.”.

We thank you for the revie

---

## [Editor Report · Decision Letter 2]

24 Jul 2025

Comparison of sequential data analysis and functional data analysis for locomotor adaptation

PONE-D-24-38528R2

Dear Dr. Quinlivan,

We’re pleased to inform you that your manuscript has been judged scientifically suitable for publication and will be formally accepted for publication once it meets all outstanding technical requirements.

Kind regards,

Gennady S. Cymbalyuk, Ph.D.

Academic Editor

PLOS ONE
---

## [Editor Report · Acceptance letter]

PONE-D-24-38528R2

PLOS ONE

Dear Dr. Quinlivan,

I'm pleased to inform you that your manuscript has been deemed suitable for publication in PLOS ONE. Congratulations! Your manuscript is now being handed over to our production team.

Kind regards,

on behalf of

Dr. Gennady S. Cymbalyuk

Academic Editor

PLOS ONE